# Retinal Microvasculature Changes Linked to Executive Function Impairment after COVID-19

**DOI:** 10.3390/jcm13195671

**Published:** 2024-09-24

**Authors:** Mar Ariza, Barbara Delas, Beatriz Rodriguez, Beatriz De Frutos, Neus Cano, Bàrbara Segura, Cristian Barrué, Javier Bejar, Mouafk Asaad, Claudio Ulises Cortés, Carme Junqué, Maite Garolera

**Affiliations:** 1Grup de Recerca en Cervell, Cognició i Conducta, Consorci Sanitari de Terrassa (CST)-Hospital Universitari, 08227 Terrassa, Spain; mariza@cst.cat (M.A.); ncanom@cst.cat (N.C.); 2Unitat de Psicologia Mèdica, Departament de Medicina, Universitat de Barcelona (UB), 08036 Barcelona, Spain; 3Ophtalmology Department, Consorci Sanitari de Terrassa (CST)-Hospital Universitari, 08227 Terrassa, Spain; bdelas@cst.cat (B.D.); brodriguez@cst.cat (B.R.); bdefrutos@cst.cat (B.D.F.);; 4Departament de Ciències Bàsiques, Universitat Internacional de Catalunya, 08195 Sant Cugat del Vallès, Spain; 5Institut d’Investigacions Biomèdiques August Pi i Sunyer (IDIBAPS), 08036 Barcelona, Spain; 6Institut de Neurociències, Universitat de Barcelona (UB), 08035 Barcelona, Spain; 7Departament de Ciències de la Computació, Universitat Politècnica de Catalunya-BarcelonaTech, 08034 Barcelona, Spainia@cs.upc.es (C.U.C.); 8Neuropsychology Unit, Consorci Sanitari de Terrassa (CST)-Hospital Universitari, 08227 Terrassa, Spain

**Keywords:** post-COVID-19 condition, optical coherence tomography angiography, retina, vessel density, superficial capillary plexus, executive function, cognition

## Abstract

**Background/Objectives**: Studies using optical coherence tomography angiography (OCTA) have revealed that individuals recovering from COVID-19 have a reduced retinal vascular density (VD) and larger foveal avascular zones (FAZs) than healthy individuals, with more severe cases showing greater reductions. We aimed to examine aspects of the retinal microvascularization in patients with post-COVID-19 condition (PCC) classified by COVID-19 severity and how these aspects relate to cognitive performance. **Methods**: This observational cross-sectional study included 104 PCC participants from the NAUTILUS Project, divided into severe (n = 59) and mild (n = 45) COVID-19 groups. Participants underwent cognitive assessments and OCTA to measure VD and perfusion density (PD) in the superficial capillary plexus (SVP) and FAZ. Analysis of covariance and partial Pearson and Spearman correlations were used to study intergroup differences and the relationships between cognitive and OCTA variables. **Results**: Severe PCC participants had significantly lower central (*p* = 0.03) and total (*p* = 0.03) VD, lower central (*p* = 0.02) PD measurements, and larger FAZ areas (*p* = 0.02) and perimeters (*p* = 0.02) than mild cases. Severe cases showed more cognitive impairment, particularly in speed processing (*p* = 0.003) and executive functions (*p* = 0.03). Lower central VD, lower central PD, and larger FAZ areas and perimeters were associated with worse executive function performance in the entire PCC sample and in the mild COVID-19 group. **Conclusions**: Retinal microvascular alterations, characterized by reduced VD and PD in the SVP and larger FAZ areas, were associated with cognitive impairments in PCC individuals. These findings suggest that severe COVID-19 leads to long-lasting microvascular damage, impacting retinal and cognitive health.

## 1. Introduction

Post-coronavirus disease 2019 condition (PCC) refers to the emergence of new symptoms or the persistence of existing symptoms three months after the first SARS-CoV-2 infection. To be considered PCC, the symptoms should persist for at least two months and cannot be attributed to any alternative cause [1]. Typical symptoms encompass fatigue, cognitive impairment, respiratory challenges, manifestations of depression and anxiety, and alterations in the senses of smell and taste. The prevalence of PCC has not yet been confirmed because its occurrence fluctuates among various study groups, research methods, SARS-CoV-2 mutations, and immunization statuses. However, according to recent estimates, approximately 12.5% of those who contract COVID-19 will develop PCC [2,3].

Meta-analyses have indicated that 7.2% to 59.2% of individuals who recover from COVID-19 develop post-COVID-19 cognitive dysfunction (PCCD) [4]. PCCD is a novel cognitive disability with a characteristic profile of executive function, memory, and attention–processing speed deficits [4,5,6]. Research suggests that these cognitive impairments generally diminish over time but can continue for more than 12 months after the infection, particularly in cases of severe initial disease [4]. Factors associated with PCCD are the severity of COVID-19 [7,8,9], age [5,8,10], sex [7,8], and the presence of comorbidities, particularly obesity [8,11].

Optical coherence tomography angiography (OCTA) is an imaging technique used to study blood flow in the eye’s vascular structures in 3D, without requiring the intravenous administration of fluorescent dyes. It detects differences between sequential B-scans to visualize blood flow. This process is repeated across different retinal positions to create a 3D dataset, which is then reconstructed into angiograms using specialized algorithms [12]. OCTA devices primarily assess vessel density (VD) and microcirculation features in two distinct retinochoroidal layers: the superficial vascular plexus (SVP) and the deep capillary plexus (DCP). The SVP provides a blood supply to the ganglion cell layer (GCL) and a portion of the retina’s inner plexiform layer, whereas the DCP supplies blood to a part of the inner nuclear layer and the outer plexiform layer in the retina [13].

Studies using OCTA conducted 15 to 45 days after the initial phase of the disease revealed that individuals recovering from COVID-19 had reduced vascular density (VD) in the central and surrounding parts of the retina, as well as in more extensive foveal avascular zone (FAZ) areas, compared to healthy individuals [14,15]. These findings have been consistently upheld in investigations conducted over a medium- to long-term duration. A recent study that involved a sample of 338 patients with COVID-19, who had been hospitalized for 4 months post-infection, and 49 healthy persons discovered a decrease in VD in all retinal plexuses of the COVID-19 group. The authors found that oxygen therapy reduced the macular vascular parameters, but the severity of the disease was not related to the vascular parameters [16]. Furthermore, a decrease in VD in the superior, nasal, inferior, and temporal areas of the SVP [17,18,19] and an increase in the FAZ area in the SVP [17,19] compared to healthy controls was demonstrated in individuals 6 months after the onset of COVID-19.

The severity of COVID-19 is linked to long-lasting changes in the small blood vessels of the retina, specifically a decrease in VD and an increase in the size of the FAZ [20,21,22]. In their investigation of 96 patients with COVID-19 (21 intensive care unit [ICU], 24 hospitalized, and 24 mild cases), Zapata et al. found there was a drop in VD in the macula 70 days after infection. The reduction in VD was associated with the severity of the disease, with more severe cases showing larger reductions [20]. A subsequent investigation 8 months later indicated that the superficial VD in an area concentrated around the fovea continued to be significantly lower in severe and moderate COVID-19 cases than that of mild cases and controls. Additionally, severe cases exhibited a wider FAZ [21]. Other authors have reported differences between severity categories for DCP but not SVP. Specifically, those with severe COVID-19 were seen to have significantly lower VD than mild individuals or healthy controls [23]. Finally, research conducted on a group of predominantly young women with PCC who experienced minor COVID-19 and did not have any retinal microvasculature-related health problems revealed no significant differences in OCTA measurements when compared to a group of healthy individuals [24].

The retina is an extension of the CNS. The small blood vessels of the retina and brain share embryological origins and are anatomically and physiologically similar. Poor functioning of the blood–brain and blood–retinal barriers is associated with the progression of retinal and cerebral microangiopathy [25,26]. Changes in retinal microvasculature are linked to the presence and progression of cerebral small-vessel disease and can serve as indicators for stroke onset and mortality [27,28,29].

According to Yeh et al.’s meta-analysis, the VD of the SVP was significantly lower while the FAZ was significantly enlarged in Alzheimer’s disease (AD) and mild cognitive impairment (MCI) individuals compared to those of the controls [30]. Similarly, research on several populations has revealed that retinal microvascular changes are related to cognitive measures. Positive correlations between SVP VD and global cognition [31,32], executive function [32,33], and memory and attention–processing speed [33] have been found in healthy older adults. The parafoveal VD of the SVP has been shown to have a positive correlation with global cognition scores in a sample of amnestic MCI and AD cases [34,35]. Additionally, among patients with chronic kidney disease, the VD of the DCP was determined to be associated with early cognitive impairment [36]. Moreover, correlations between VD in the SVP and executive function–processing speed and hippocampal volume have been demonstrated in elderly adults [37].

To our knowledge, no studies have examined the relationships between retinal vessel density and cognitive function in PCC individuals. The present study aimed to describe the alterations in retinal microvasculature that occur with COVID-19 and their relationship with the severity of the disease and cognitive performance in people with PCC.

## 2. Materials and Methods

### 2.1. Participants

This was an observational cross-sectional study that consecutively evaluated PCC participants from the NAUTILUS Project (ClinicalTrials.gov ID: NCT05307575) at the Ophthalmology Department of the Consorci Sanitari de Terrassa-Hospital Universitari (Terrassa, Barcelona, Spain) from January 2022 to February 2024.

The inclusion criteria were as follows: (a) those with a confirmed diagnosis of COVID-19 according to WHO criteria and signs and symptoms of the disease presenting during the acute phase; (b) cases at least 12 weeks after infection; and (c) patients aged 18–65 years. The exclusion criteria were as follows: those with (a) an established diagnosis of psychiatric, neurological, or neurodevelopmental disorders or systemic pathologies known to cause cognitive deficits before the episode of COVID-19; (b) motor or sensory alterations that impede neuropsychological examination; (c) type 1 or 2 diabetes; (d) a history of intraocular or refractive surgery; (e) an absolute spherical refractive error of > 5 diopters or axial length of > 26 mm; (f) glaucoma or any retinal disease; and (g) corrected visual acuity of less than 20/25. All participants were native Spanish speakers.

The sample size was determined using the VD as the primary outcome. The computation was performed using G*Power V3.1.9.6.1 software [38]. To meet the sample size requirements of the power analysis with a β-error of 95% and a significance level of 5%, the minimum sample size recommended was 42 participants for each group. Based on previous research [20,21], we estimate the effect size of the differences in VD measures between people with severe and mild COVID-19 to be moderate to large.

The study was conducted with the approval of the Drug Research Ethics Committee (CEIm) of Consorci Sanitari de Terrassa (CEIm code: 02-20-107-070) and the Ethics Committee of the University of Barcelona (IRB00003099).

### 2.2. Procedure

The entire process comprised three sessions. During the initial session, we collected data on demographic characteristics, prior comorbidities (heart disease, respiratory disease, high blood pressure, dyslipidemia, obesity, thyroid disease, chronic liver disease, tobacco smoking), and COVID-19 and post-COVID-19 symptoms. Each participant underwent a cognitive assessment in the second session with a comprehensive neuropsychological battery, as described by Ariza et al. [6]. Briefly, for this study, we used the following neuropsychological tests: parts A and B of the Trail-Making Test (TMT) for visual scanning, motor speed and attention, and mental flexibility [39], with the difference in scores (B-A) determined to remove the speed element from the test evaluation [40]; the WAIS-IV Digit Span Subtest, which was used to measure verbal attention (Digits Forward) and working memory (Digits Backward) [41]; the Controlled Oral Word Association test (COWAT) [42] for phonetic fluency; the Digit Symbol Coding Test (WAIS-III) [41] for visual scanning, tracking, and motor speed; and the Stroop Color and Word test (SCWT) [43], which measures processing speed (SCWT Word and Color) and cognitive inhibitory control (SCWT Word-Color). These instruments are recommended for evaluating post-COVID-19 cognitive impairment [44], and previous studies have shown that these instruments are responsive to alterations in executive function and mental processing speed [9,45,46]. Finally, we used the Montreal Cognitive Assessment (MoCA) for general cognitive screening [47,48]. We classed a cognitive deficit as present if one of the neuropsychological subtests was less than −1.5 SD or if two subtests of the same cognitive domain were less than 1 SD below the mean. Participants were classified as cognitively impaired if they presented a deficit in at least two cognitive domains [49].

In the third session, all participants underwent a complete ocular examination that included a measurement of corrected visual acuity, an intraocular pressure assessment using a contact tonometer, biomicroscopic and funduscopic exams after dilation with topical tropicamide, and axial length determination through an optical biometer (IOLMaster 500, Carl Zeiss Meditec Inc., Dublin, CA, USA).

Structural optical coherence tomography (OCT) and OCTA were performed for the macula (Cirrus HD-OCT 5000, Carl Zeiss Meditec Inc., Dublin, CA, USA). Macular OCTA images of 3.0 × 3.0 centering measurements on the foveal area (mm^2^) were acquired. The OCT device automatically segmented the SVP and generated data encompassing VD and PD for the central (vessels in the central area of the macula around the fovea), internal (surrounding the central region but excluding the peripheral extension), and complete (integrating the central and peripheral regions) SVP and metrics for the FAZ, including area, perimeter, and circularity (Figure 1). To maintain data independence, simplify the analysis, and avoid duplicating sample size, only one eye was chosen for each participant, and we selected the eye with the best image quality. The examiner who performed was blinded to the group attribution.

### 2.3. Statistical Analyses

Descriptive statistics were obtained for all study variables. Graphical representations and descriptive statistics were used to study the assumptions (Shapiro–Wilk test for normality and the Levene test for homogeneity of variances). Demographic differences between groups were examined using Student’s *t*-test and the U Mann–Whitney test for quantitative measures and Chi2, the Fisher exact test, and the independent-samples proportions test for categorical measures. To study the differences in study variables between the groups, analysis of covariance (ANCOVA) was performed. The effect size was calculated using the partial eta squared value (ή_p_^2^). Partial Pearson or Spearman correlations were performed between the OCTA and cognitive variables. All analyses were corrected for confounding variables when these variables were distributed differently among groups. Analyses were performed using IBM SPSS Statistics 29.0 (IBM Corp., Armonk, NY, USA). The level for statistical significance was set at α = 0.05.

## 3. Results

### 3.1. Demographic and Clinical Characteristics

One hundred four participants with PCC were recruited. In accordance with the WHO COVID-19 clinical progression scale [50], they were classified into severe PCC (H-PCC) (n = 59) and mild PCC (M-PCC) (n = 45). All the hospitalized patients obtained at least 6 points on the WHO Clinical Progression Scale (non-invasive ventilation or high-flow nasal cannula) and thus met the severity criterion. Half of the severe cases required ICU management. The participants’ sociodemographic characteristics and comorbidities are shown in Table 1. The groups were equivalent in mean age, but the M-PCC group had a higher proportion of females and a higher educational level than the H-PCC group. On average, all PCC participants had shown positivity in a SARS-CoV-2 test conducted 390 days before their neuropsychological evaluation (SD = 276.45 days), and the H-PCC group had fewer days of evolution since the start of COVID-19 than the M-PCC group. Premorbid high blood pressure (HBP) and obesity were more prevalent among hospitalized participants than M-PCC participants. Therefore, for OCTA analyses, sex, high blood pressure, and time since a positive test were considered confounding variables, while for cognitive analyses, sex, educational level, and time since testing positive were controlled variables.

In Figure 2, the symptoms reported by individuals with PCC at the time of evaluation are presented as percentages. The comparison of proportions revealed that individuals with PCC-M reported a higher proportion of cognitive symptoms, fatigue, headache, dizziness, altered smell and taste, chills/flushing, diarrhea, and cognitive complaints than those with PCC-H. On the other hand, participants who had severe COVID-19 most frequently reported between 1 and 6 symptoms, while those who had mild COVID-19 mostly reported between 7 and 12 symptoms.

### 3.2. Optical Coherence Tomography Angiography Results

Table 2 displays the scores for the optical coherence tomography angiography variables adjusted for sex, time since testing positive, and presence of HBP. Choroidal thickness was significantly lower in the H-PCC group than in the M-PCC group. Central VD and PD and total VD SVP measurements were significantly lower in participants in the H-PCC group than in those in the M-PCC group. FAZ area and perimeter measurements were significantly greater for participants in the H-PCC group than for those in the M-PCC group.

### 3.3. Cognitive Results

As Table 3 shows, the H-PCC group performed significantly worse in the Digit Span Forward, Digit Symbol, and SCWT Color-Word tests after controlling for sex, educational level, and time since testing positive. While those with mild COVID-19 experienced a noticeably higher occurrence of cognitive complaints than those with severe symptoms, the H-PCC group demonstrated slightly more pronounced cognitive impairment, as indicated by the neuropsychological assessments (Figure 3). Nevertheless, these disparities did not demonstrate statistical significance.

### 3.4. Correlation Results

We analyzed the relationship between OCTA metrics and cognitive test results while considering the effects of age, schooling, and number of days from infection to evaluation for all participants (Table 4). The central VD showed a considerable inverse correlation with TMT B and the difference in time between TMT A and B (TMT B-A). CVD also directly correlated with the score in the SCWT Color-Word. The central PD inversely correlated with the TMT B and TMT B-A and directly correlated with the SCWT Color-Word score. The TMT A, TMT B, and TMT B-A directly correlated with the area and perimeter of the FAZ. Moreover, the FAZ exhibited an inverse correlation with SCWT Color-Word, for area and perimeter. We then repeated the same analyses for each group separately. The M-PCC group obtained results similar to those of the combined group but with even greater statistical significance (Table 5). However, the H-PCC group’s correlations did not reach statistical significance. Figure 4 and Figure 5 show the partial correlations between central VD and cognitive performance and FAZ and cognitive performance for the two groups and total PCC participants.

## 4. Discussion

Previous studies have demonstrated a decrease in the VD of the SVP [17,18,19] and an increase in FAZ [17,19] in individuals 6 months after the onset of COVID-19 compared to healthy controls. We compared the retinal microvasculature of PCC individuals with severe and mild COVID-19 to assess the systemic impact of the disease and its relationship with vascular damage. Participants with persistent symptoms who had been hospitalized during the acute phase (half of whom required ICU) showed worse retinal microvascularization metrics than those with mild PCC. VD (central and total) and CPD were lower in the eyes of patients with severe PCC, and their FAZ areas and perimeters were larger. These results agree with a study by Banderas-García et al. [21], who reported higher VD and larger FAZ areas in severe and moderate cases at 8 months post-COVID-19 compared to mild cases and controls.

Severe and mild cases of COVID-19 are differentiated based on clinical criteria such as the severity of respiratory symptoms, the need for supplemental oxygen, and the presence of complications. Severe cases have more intense symptoms, require hospitalization and oxygen therapy, and often have complications such as severe pneumonia and hypoxia [50]. These patients exhibit a strong inflammatory response, an increased risk of thrombosis, and generalized endothelial dysfunction [51]. In contrast, mild cases usually present moderate symptoms, such as fever, cough, and loss of smell or taste, and do not require hospitalization. These patients have less systemic inflammation, a lower risk of blood clots, and less endothelial dysfunction.

The occurrence of microvascular damage in severe COVID-19 is due to several interrelated mechanisms: cytotoxic damage to endothelial cells by the virus [52], a hypercoagulable state that produces microthrombi in small vessels [53], cytokine storm [54], and excessive activation of the complement system [55]. Complement activation is a distinctive characteristic of SARS-CoV-2 infection that is missing from other viral infections. Certain markers of complement activation are associated with worse outcomes, including an increased risk of ICU admission and the need for invasive mechanical ventilation, in patients with COVID-19 [55].

We studied a cohort characterized by persistent symptoms for more than three months and compared the groups sorted according to the severity of the disease. Although PCC appears to be more prevalent in severe COVID-19 cases [56,57], many mild cases also develop PCC [58]. Months after recovery, people who had severe COVID-19 frequently showed antibody formation [59] and chronic inflammation [10]. People with long COVID-19 had greater complement activation during acute illness, and this activation remained unchanged after six months in cases of severe COVID-19 [60]. Excessive activation of the complement system plays critical roles in the progression of ophthalmological (age-related macular degeneration, glaucoma) [61,62] and neurodegenerative diseases (Alzheimer’s disease, amyotrophic lateral sclerosis, Huntington’s disease, and Parkinson’s disease) [63,64].

Studies looking at retinal microvasculature in people with post-COVID-19 syndrome have reported mixed results [24,65]. One study found no deterioration of the SVP in patients with PCC at an average of 15 months after infection when compared to the retinas of non-COVID-19 controls [24]. It should be noted that all but one participant in this study had mild COVID-19. However, a study by Schlick et al. [65] demonstrated that PCC individuals had impaired VD in the intermediate capillary plexus. Furthermore, when these authors compared PCC individuals, they found that the VD in the SVP of those who had fatigue as a persistent symptom was lower than that of those who did not [65].

It is unknown to what extent microvascular damage is generated during the acute phase of the disease or with persistent symptoms, but in any case, it seems to be related to the severity of the disease. Severe cases are more likely to experience pronounced alterations in retinal VD and other organ dysfunctions due to extensive microvascular damage, systemic inflammation, and hypoxia [66].

The results of the H-PCC and M-PCC groups differed in one test of attention (Digit Span Forward), one test of processing speed (Digit Symbol), and one of executive function (SCWT Color-Word). In a previous study conducted with three PCC groups (ICU, hospitalized, and mild) and a healthy control group, we demonstrated that an executive function test could differentiate between the ICU and mild groups [9]. Here, we focused on the severity of the disease and discovered that the performance in tests of mental attention–processing speed and executive function not only distinguished individuals with COVID-19 from those without the disease [6,9] but also differentiated those with severe COVID-19 from those with mild COVID-19. Moreover, we demonstrated that the integrity of the retinal microvasculature correlated with the executive function test performance of the total PCC sample and the M-PCC subgroup.

Relationships between microvascular alterations and mild cognitive impairment (MCI) or AD have been previously reported [30]. Additionally, correlations between these alterations and cognitive performance have been established. Specifically, executive function, processing speed, and memory have been linked to SVP VD in healthy older adults [32,33,37] and to global cognition in individuals with amnesic MCI and AD [34,35]. However, to our knowledge, these relationships have not been previously reported in individuals with PCC. Although the correlations between microvascularization measures and cognitive variables are statistically significant, they show a moderate effect size. However, given the characteristics of the PCC population—mostly young adults of working age experiencing cognitive impairments—this relationship is clinically significant. The detection of microvascular alterations could serve as an early indicator of cognitive risk, warranting the referral of these patients for a comprehensive neuropsychological evaluation. Early intervention would allow for the identification of cognitive deficits that might otherwise go unnoticed, facilitating timely intervention to prevent progressive cognitive decline or the development of dementia in the future.

The SVP, located in the GCL, is crucial for the metabolism of ganglion cells [13], and its thinning is observed in neurodegenerative diseases [67]. Changes in brain regions responsible for visual processing can cause retrograde degeneration in the retina, reflecting SVP dysfunction [68]. Moreover, SVP dysfunction may cause anterograde degeneration and thus thinning of the retinal layers, leading to changes in the gray matter microstructure related to vision [69]. Neuroimaging studies in healthy adults have linked poor microvascular perfusion in the SVP to reduced gray matter volume in areas of the occipital lobe and structures involved in visual processing such as the primary visual cortex, lingual, temporal, and fusiform gyri [70]. Additionally, a reduced SVP has been associated with less gray matter in the parahippocampal gyrus, anterior cingulate cortex, and medial temporal gyrus [70]. Dysfunctions of the primary and secondary visual areas can affect connectivity with the prefrontal cortex, interfering with the information flow necessary for executing complex functions. These areas are crucial for decision-making, planning, perception, and visual attention, and their proper functioning is essential for the visual feedback needed for inhibitory control and error correction [71]. The relationships between retinal microvascular alterations and executive function deficits found in this study are consistent with the above functional descriptions. Future neuroimaging studies in people with PCC will help identify whether there are relationships between brain changes, cognitive impairments, and reduced retinal VD.

It was unexpected that the correlations between OCTA metrics and neuropsychological tests in the H-PCC group were not statistically significant. However, there are several possible explanations for these results. First, compared with the mild and combined PCC groups, the group of patients with severe PCC may have had greater heterogeneity in the severity of symptoms and complications. This heterogeneity could have masked the relationships between the OCTA metrics and cognitive variables. Furthermore, the patients with severe PCC could have had other medical complications that affected their retinal and cognitive functions, thus confounding and diluting the observed correlations. Second, in cases of extensive damage, the OCTA metrics and cognitive variables could have approached a damage saturation threshold, such that additional differences are not clearly reflected in the correlations. If the OCTA metrics and cognitive functions were already severely impaired, there may have been a ceiling or floor effect that limited the variability and, therefore, the observed correlations. The absence of notable correlations in the severe group suggests that additional factors that interfere with the detection of these correlations can affect patients with severe PCC. Exploring these factors and adjusting for within-group heterogeneity could provide a deeper understanding of the dynamics underlying these observations.

Our study has several limitations. Although the number of participants per group was sufficient to detect differences between them, the selected sample may not fully represent the population of people with PCC. The cross-sectional nature of our study design prevented us from determining causal relationships with precision. Additionally, we did not have data on the pre-infection status of the eyes of the study participants; therefore, we cannot rule out the possibility that some of these lesions were present before COVID-19 infection. Moreover, our severity groups were heterogeneous in terms of their demographic variables (sex and level of education), comorbidities (hypertension and obesity), and time from infection to evaluation. Some of these variations could not be avoided but are risk factors for more or less severe forms of COVID-19 or the development of PCC. However, we considered these differences and used statistical correction to avoid bias when comparing the two groups. Finally, we did not study the cerebral vasculature of the patients.

We focused on the SVP because it offers a clear view of the capillary network near the retinal surface, making it useful for detecting early perfusion changes and vascular alterations. It is widely studied, allowing for easier comparison with existing data. In contrast, overlapping structures make deep plexus images, which provide detailed microvascular information for understanding extensive post-COVID-19 damage, more difficult to interpret. Additionally, fewer studies have focused on the deep plexus for post-COVID-19 retinal assessment, making it difficult to compare and contextualize the results. Despite these limitations, to our knowledge, this is the first study to provide a report on the relationship between retinal microvasculature and cognitive impairment in people with PCC. Moreover, unlike the patients in other studies [32,72], these participants were cognitively explored in person with an extensive neuropsychological battery commonly used in the clinical context, which validated its applicability. Furthermore, sample selection was carried out by ruling out comorbidities that could cause cognitive impairment.

## 5. Conclusions

Our results demonstrated that people with post-COVID-19 condition who previously had severe COVID-19 had lower central vascular density and perfusion density in the superficial capillary plexus and larger foveal avascular zone areas and perimeters than those who had experienced a mild form of the disease. Furthermore, these alterations in retinal microvasculature were related to the patients’ neuropsychological performance, particularly their executive functions. These findings suggest that severe COVID-19 leads to long-lasting microvascular damage that impacts retinal and cognitive health. Follow-up studies, including brain neuroimaging techniques, are needed to explore these relationships and their implications for post-COVID-19 rehabilitation strategies.

## Figures and Tables

**Figure 1 jcm-13-05671-f001:**
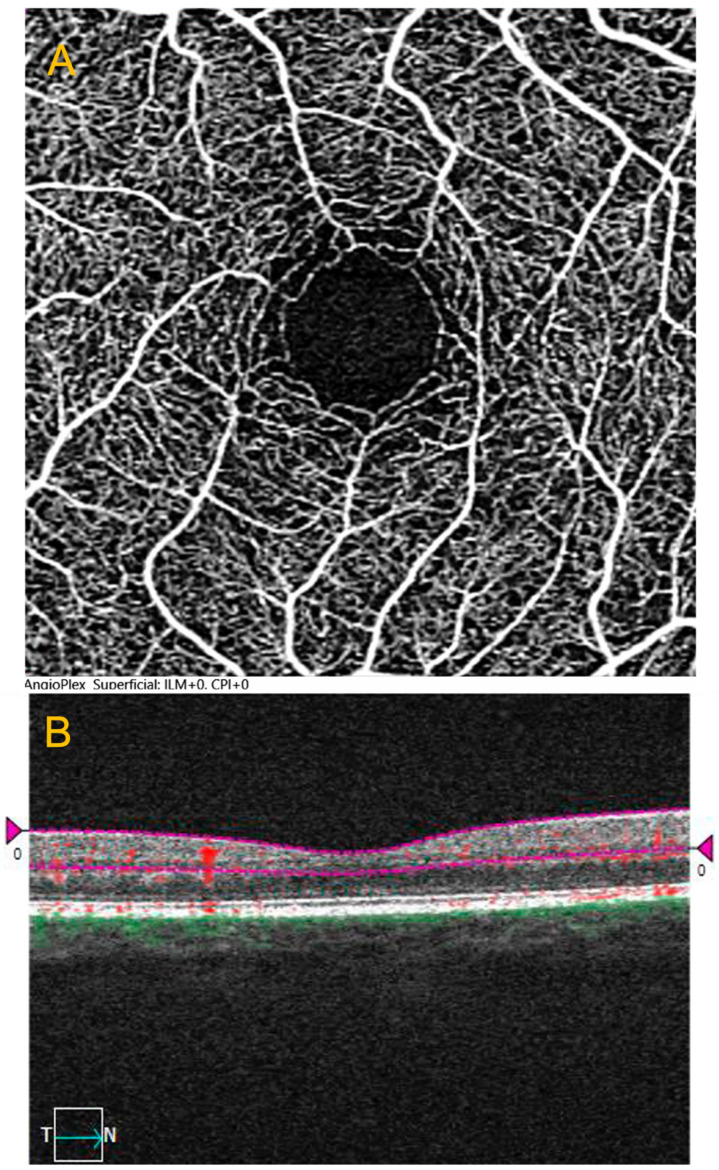
OCTA images: (**A**) 3 × 3 mm en face optical coherence tomography angiography (OCTA) of the superficial capillary plexus (SVP); (**B**) OCT B-scan with angioflow (shown in red).

**Figure 2 jcm-13-05671-f002:**
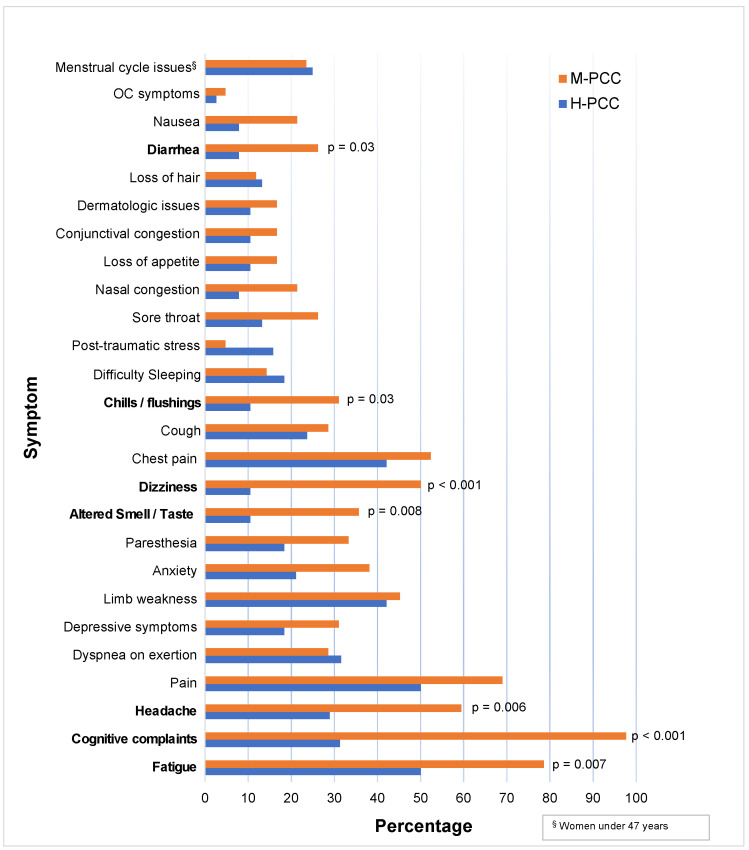
Symptoms reported by people with PCC at the time of assessment according to the severity of COVID-19 expressed as percentages. The symptoms whose proportions differ statistically significantly between the groups are highlighted in bold. The associated *p*-values are presented. OC = obsessive–compulsive. ^§^ The percentage of menstrual cycle issues was computed for the sample of women under 47 years of age.

**Figure 3 jcm-13-05671-f003:**
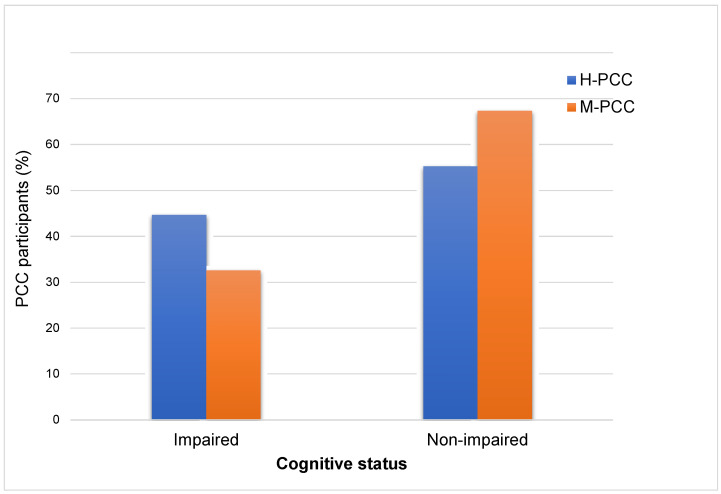
Cognitive status according to the severity of COVID-19. The graph illustrates the % of participants with PCC who had cognitive impairments.

**Figure 4 jcm-13-05671-f004:**
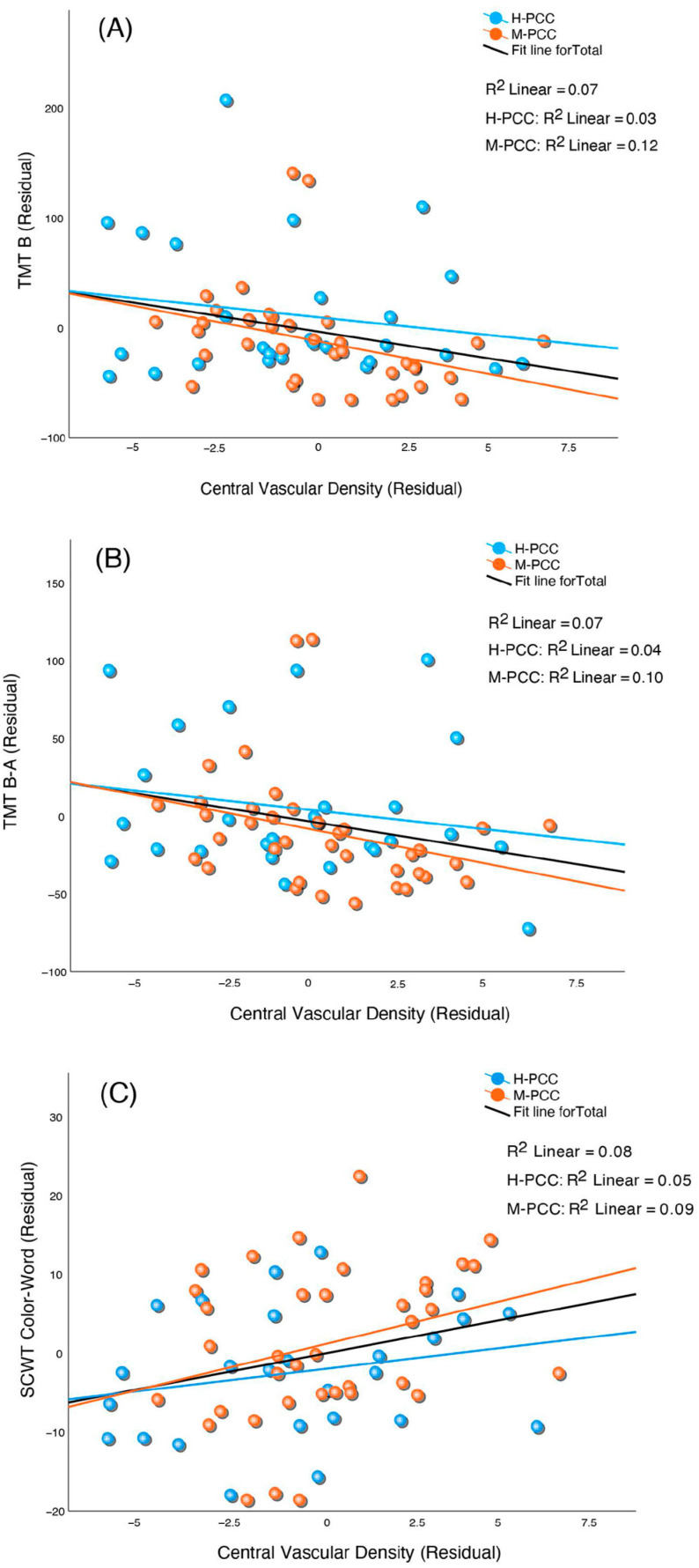
Grouped scatter plots showing the partial correlation between central vascular density (CVD) of the superficial capillary plexus (SVP) and cognitive performance measures: (**A**) Trail-Making Test (TMT) part B, (**B**) TMT part B minus part A, and (**C**) Stroop Color and Word test (SCWT) Color-Word. The relationships are depicted for PCC severity groups (mild and severe) and the combined group. Partial correlations controlled for age, years of schooling, and time from infection to assessment.

**Figure 5 jcm-13-05671-f005:**
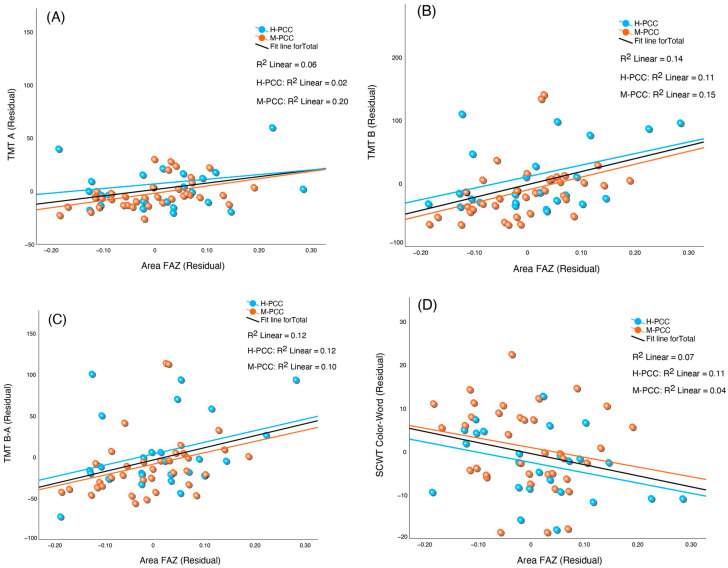
Grouped scatter plots showing the partial correlation between foveal avascular zone (FAZ) and cognitive performance measures: (**A**) Trail-Making Test (TMT) part A, (**B**) TMT part B, (**C**) TMT part B minus part A, and (**D**) Stroop Color and Word test (SCWT) Color-Word. The relationships are depicted for PCC severity groups (mild and severe) and the combined group. Partial correlations controlled for age, years of schooling, and time from infection to assessment.

**Table 1 jcm-13-05671-t001:** Sociodemographic characteristics and comorbidities of the PCC severity groups.

	H-PCC n = 59	M-PCC n = 45	
	Mean (SD)	Mean (SD)	*p*-Value
Age	51.64 (6.52)	50.32 (8.16)	0.36
Education (years)	12.84 (3.32)	14.45 (3.79)	**0.05**
Evolution (months) ^†^	6 (5.8–11.45)	14.81 (10.64–24.26)	**<0.001**
Axial length (mm)	23.95 (1.18)	23.48 (0.88)	**0.03**
Corrected visual acuity	0.975 (0.066)	0.970 (0.803)	0.39
	**N (%)**	**N (%)**	***p*-value**
Sex (% female)	25 (39.1)	39 (60.9)	**<0.001**
**Comorbidities**			
*Heart disease*	1 (1.7)	1 (2.3)	
*Respiratory disease*	13 (22.4)	8 (18.2)	0.63
*High blood pressure*	13 (22.4)	2 (4.5)	**0.01**
*Dyslipidemia*	9 (15.5)	5 (11.4)	0.77
*Obesity*	19 (32.8)	4 (9.1)	**0.005**
*Thyroid disease*	5 (8.6)	3 (6.8)	0.74
*Chronic liver disease*	5 (8.6)	0	
*Tobacco smoking*	5 (8.6)	5 (11.4)	0.65
**ICU management**	29 (50)		
*Endotracheal intubation*	9 (31)		
*High-flow oxygen therapy*	18 (62)		
*Non-invasive mechanical ventilation*	9 (31)		
*Corticosteroids (low doses)*	20 (68.9)		
*Corticosteroids (high doses)*	2 (6.89)		
*Dexmedetomidine*	8 (27.5)		
*Hydroxychloroquine*	5 (17.24)		
*Azithromycin*	4 (13.8)		
*Tocilizumab*	8 (27.5)		
*Ceftriaxone*	8 (27.5)		

PCC = post-COVID-19 condition; H = severe; M = mild; ^†^ evolution is the time since the first positive test (months) to assessment. Data presented as the median and interquartile range (Q1–Q3). Mann–Whitney test (U Statistic).

**Table 2 jcm-13-05671-t002:** PCC severity groups’ mean differences in optical coherence tomography angiography measures adjusted for sex, time since positive test, and HBP status.

	H-PCC n = 59	M-PCC n = 45			
	M_adj_ (SE)	M_adj_ (SE)	F	*p*-Value	η_p_^2^
Macular thickness (µm)	258.71 (4.82)	268.65 (3.34)	2.552	0.12	0.044
Choroidal thickness (µm)	249.24 (12.58)	312.39 (11.61)	11.360	**0.001**	0.140
Central VD	10.74 (0.65)	13.07 (0.68)	5.127	**0.03**	0.064
Internal VD ^†^	21.75 (19.10–22.73)	22.20 (21.48–22.83)	0.869	0.35	
Full VD	19.38 (0.42)	20.82 (0.44)	4.686	**0.03**	0.059
Central PD (%)	17.84 (1.06)	21.65 (0.94)	6.246	**0.02**	0.092
Internal PD (%) ^†^	37.80 (0.66)	39.76 (0.70)	0.941	0.34	
Full PD (%) ^†^	35.43 (0.63)	37.71 (0.67)	2.203	0.14	
FAZ area (mm^2^)	0.284 (0.018)	0.218 (0.019)	5.329	**0.02**	0.067
FAZ perimeter (mm) ^†^	2.388 (0.086)	2.065 (0.090)	5.661	**0.02**	0.071

HBP = high blood pressure; VD = vascular density; PD = perfusion density; FAZ = foveal avascular zone; ^†^ data presented as the median and interquartile range (Q1–Q3). Quade non-parametric ANCOVA; η^2^ effect size is as follows: η^2^ = 0.009, small; η^2^ = 0.059, medium; η^2^ = 0.139, large; all OCTA measurements were acquired in the superficial capillary plexus.

**Table 3 jcm-13-05671-t003:** PCC severity groups’ mean differences in cognitive measures adjusted for education, sex, and time since a positive test.

	H-PCC n = 59	M-PCC n = 45			
	M_adj_ (SE)	M_adj_ (SE)	F	*p*-Value	η_p_^2^
MoCA ^†^	26 (24.5–27)	27 (24–28.5)	1.393	0.24	
Digit Span Forward (raw score)	5.07 (0.26)	5.92 (0.24)	5.010	**0.03**	0.063
Digit Span Backward (raw score) ^†^	4 (3–5)	4 (4–5)	0.172	0.68	
Digit Symbol (raw score)	54.12 (3.00)	67.56 (2.79)	9.424	**0.003**	0.113
TMT A (time in s) ^†^	38 (22–51.50)	33 (26.75 –)	1.127	0.26	
TMT B (time in s) ^†^	80 (53.50–143)	72.50 (53.75–95)	1.957	0.05	
TMT B-A (time in s) ^†^	43 (25–73)	34 (22.75–57)	2.544	0.12	
SCWT Word (raw score)	88.63 (3.72)	92.05 (3.34)	0.403	0.53	0.006
SCWT Color (raw score)	60.05 (2.86)	63.65 (2.57)	0.750	0.39	0.010
SCWT Color-Word (raw score)	33.52 (1.73)	38.91 (1.58)	4.672	**0.033**	0.060
COWAT (sum raw score)	36.63 (2.13)	40.49 (2.04)	1.497	0.23	0.020

PCC = post-COVID-19 condition; H = severe; M = mild; MoCA = Montreal Cognitive Assessment; TMT = Trail-Making Test; SCWT = Stroop Color and Word test; COWAT = Controlled Oral Word Association Test. ^†^ Data presented as the median and interquartile range (Q1–Q3). Quade non-parametric ANCOVA; η^2^ effect size is as follows: η^2^ = 0.009, small; η^2^ = 0.059, medium; η^2^ = 0.139, large.

**Table 4 jcm-13-05671-t004:** Statistically significant correlations between optical coherence tomography angiography metrics and cognitive variables controlled for age, years of education, and time from infection to assessment for all PCC participants.

OCTA Metric	Cognitive Variable	Pearson r Partial	*p*	Spearman rs Partial	*p*
CVD	TMT B			−0.324	**0.009**
	TMT B-A			−0.333	**0.007**
	SCWT CW	0.321	0.020		
CPD	TMT B			−0.303	**0.02**
	TMT B-A			−0.313	**0.01**
	SCWT CW	0.323	0.019		
FAZ Area	TMT A			0.338	**0.007**
	TMT B			0.420	**<0.001**
	TMT B-A			0.381	**0.002**
	SCWT CW	−0.284	0.041		
FAZ Perimeter	TMT A			0.266	**0.04**
	TMT B			0.387	**0.002**
	TMT B-A			0.369	**0.003**
	SCWT CW	−0.309	0.026		

**Table 5 jcm-13-05671-t005:** Statistically significant correlations between optical coherence tomography angiography metrics and cognitive variables controlled for age, years of education, and time from infection to assessment for M-PCC participants.

OCTA Metric	Cognitive Variable	Spearman rs Partial	*p*
CVD	TMT A	−0.427	**0.008**
	TMT B	−0.520	**<0.001**
	TMT B-A	−0.487	**0.002**
CPD	TMT A	−0.414	**0.01**
	TMT B	−0.494	**0.002**
	TMT B-A	−0.464	**0.004**
FAZ Area	TMT A	0.545	**<0.001**
	TMT B	0.501	**0.002**
	TMT B-A	0.408	**0.01**
FAZ Perimeter	TMT A	0.559	**<0.001**
	TMT B	0.549	**<0.001**
	TMT B-A	0.455	**0.005**

## Data Availability

The data that support the findings of this study are available on request from the corresponding author. The data are not publicly available due to privacy or ethical restrictions.

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
