# Peer review of "Retinal Microvasculature Changes Linked to Executive Function Impairment after COVID-19"

_jcm, 2024, doi:10.3390/jcm13195671_

Round 1
Reviewer 1 Report
Comments and Suggestions for Authors
Retinal microvasculature changes linked to executive function impairment after COVID-19
Overview
This cross-sectional study investigated the impact of COVID-19 on retinal microvascularization. Patient groups, classified by post-COVID condition (PCC) severity as severe and mild, were compared with respect to both cognitive assessments and OCTA to measure vascular density (VD) and perfusion density (PD) in the superficial capillary plexus (SVP) and foveal avascular zones (FAZs). Severe participants had significantly lower central and total VD, lower central PD measurements, and larger FAZ areas and perimeters than mild cases. Severe cases showed more cognitive impairment, particularly in speed 34 processing and executive functions. Retinal microvascular alterations, characterized by 37 reduced VD and PD in the SVP and larger FAZ areas, were associated with cognitive impairment. These findings suggest that severe COVID-19 leads to long-lasting microvascular damage, impacting retinal and cognitive health.
Critique
This well-organized article addresses an issue that is timely. Sections are well-developed. There is a clear objective and the approaches taken address the objective well. The methodology is explained clearly. The analysis is well done. The conclusions are convincing.
Specific comments
At line 149, the authors give details of the sample size calculations except that they omit the variable on which the calculations were based. They need to state the primary outcome variable for which the sample size estimate was computed. They only say that they expect the effect size of all VD measures to be moderate to large.
In line 199, the authors state that patients were classified into severe PCC (H-PCC) (n = 59) and mild PCC (M-PCC) (n = 45). All tables and figures use these classification labels except Table 2. There they are labeled as “PCC severe” and “PCC mild.” Also, the results section for this table refers to H-PCC and M-PCC, but the columns are not labeled that way. I suggest making Table 2 column labels match those in the other tables and the text in the results.
In the statistical analysis section, the authors state that adjustment for covariates will be carried out when necessary. In the initial results paragraph, they identify the covariates and then state that they were considered confounding when appropriate. Could they be more specific for the terms “necessary” and “appropriate?” Later, I do see that they indicate which variables were used as covariates for analysis in specific tables.
For the legend and description of Figure 2, the authors state the frequency of symptoms that the two severity groups have. There are different numbers of subjects in the two groups. Therefore, the comparison should be made on proportions, not frequency. Otherwise, the results may be misleading. In addition, the testing between the two groups would have to be based on the difference in proportions, not frequency. The comparison would be clearer if proportions were given and a statement that tests for differences were based on proportions.
Figure 3 needs to be revised. The bars are meant to compare the proportion of complaints in each group. However, all the bars are the same color (black) so it’s not clear which group is which.
In Table 4, there are several “significant” correlations, however the actual magnitudes of the correlations are not very high. That should be considered as well as significance.
Figures 4 and 5 are too small to read. It is very useful that r-squared are included (I magnified the plots to be able to see that). They are very small, indicating that the regression lines are not good fits to the data.

Reviewer 2 Report
Comments and Suggestions for Authors
I would like to congratulate the Authors for their research. The article is well-written and has an overall good readability. Statistical analysis was properly performed. The results are clearly presented and the conclusions are supported by the results. The discussion is clear and exhaustive.
I invite the Authors to share the row data as supplementary materials.
Reviewer 3 Report
Comments and Suggestions for Authors
A potentially interesting paper on retinal microvasculature changes in post-COVID-19 patients. However, the editing mistakes should be corrected before the final decision. In general, the paper should be more organized (especially the discussion section) to follow easily each part of the work. Please see my comments.
The Authors should strictly follow the journal's regulations regarding the template, please correct the reference list.
Please indicate the exact results (parameters with p-values) in the Result section of the Abstract.
Lines 63-67. Is it necessary to discuss the topic of the paper? Moreover, the introduction is pretty long, almost half of all citations are there. I believe that only the most crucial aspects should be introduced.
Line 126. PPC or PCC?
Why did you include patients only aged 30-65?
Did you include active smokers, and patients with arterial hypertension/obesity in the analysis?
Descriptions of comorbidities are needed to be defined in the methods section.
Line 188. Why?
Line 199. Should be Chi2. Were all categorical data analyzed with this test, not with the Fisher exact test? What about the continuous data, the UMW test? What about the assessment of data normality?
Statistically significant p-values should be presented rounded to the third decimal place (in the case of very small values, p<0.001), and those with a value > 0.05 to the second decimal place. The t-score can be omitted.
Figure 2. Please show the exact p-values.
Line 238. Please add the full term. Also in descriptions of tables, the beginning of text in Table 2 (lines 245-246).
I believe that continuous data with normal distribution should be presented as mean with SD, and compared using Student's t-test, whereas without normality of data with Mann-Whitney test, and presented as median with Q1-Q3 ranges. P-values should be given since it is the most valid aspect.
Please bold/mark the statistically significant results in tables/figures.
Figure 4. A better resolution or bigger pictures should be presented.
At the beginning of the discussion, it would be better to analyze the main results, but it is not a place to repeat the aim of this study (it was stated previously).
Line 353. References?
It would be better not to use abbreviated forms in the conclusions since you summarize the most relevant aspects.
What about the follow-up of analyzed patients? Do you plan to explore the results in the future?
Comments on the Quality of English LanguagePlease look for typos. Grammar needs to be corrected. Also, editing problems, SARS-Cov2, etc.
Round 2
Reviewer 3 Report
Comments and Suggestions for Authors
The Authors have addressed my suggestions and comments. Thank you.
After reviewing the paper, please see my additional thoughts.
Abstract. p-value should start with 0, e.g., p = 0.026. The same problem with the whole manuscript later, and Tables.
Please do not use F statistics in the abstract. Compare 2 groups with exact values (median and Q1-Q3 ranges) and p-values as a difference.
Line 199. Should be: Chi2
Line 206. Critical level?
T and x2 statistics are not necessary to be presented in the text, p-values are enough
Figure 3. Please add %
The R-value should be rounded to the second decimal place.
Comments on the Quality of English LanguagePlease look for typos. Thank you.
Author Response
I appreciate that you have carefully reread our manuscript. I have made the changes you suggested. However, there are a couple of things I do not quite understand. Would you be so kind as to clarify this? (see answer 2 and 3)
Changes to the revised manuscript are in red.
1. Abstract. p-value should start with 0, e.g., p = 0.026. The same problem with the whole manuscript later, and Tables.
We have made the changes proposed by the reviewer in the text, tables and figure 2.
2. Please do not use F statistics in the abstract. Compare 2 groups with exact values (median and Q1-Q3 ranges) and p-values as a difference.
The F statistics have been removed from the abstract. Regarding the comment about comparing the two groups with exact values ​​(median and Q1-Q3 ranges) and p-values ​​as difference, I am sorry, I do not understand what you mean.
3. Line 199. Should be: Chi2
I do not understand what you are referring to either. It already says Chi2... although it's not on line 199.
4. Line 206. Critical level?
Following the reviewer's recommendation, we have removed the word “critical”.
5. T and x2 statistics are not necessary to be presented in the text, p-values are enough
Should I understand that the reviewer is referring to the tables? Following his recommendation, we have removed the t i x2 statistics from the tables.
6. Figure 3. Please add %
In the figure caption, we have added % (line 275)
7. The R-value should be rounded to the second decimal place.
The R2 values in Figures 4 and 5 have been rounded to the second decimal place.